# Adhering to Canada’s Food Guide Recommendations on Healthy Food Choices Increases the Daily Diet Cost: Insights from the PREDISE Study

**DOI:** 10.3390/nu14183818

**Published:** 2022-09-16

**Authors:** Gabrielle Rochefort, Didier Brassard, Marie-Claude Paquette, Julie Robitaille, Simone Lemieux, Véronique Provencher, Benoît Lamarche

**Affiliations:** 1Centre Nutrition, Santé et Société (NUTRISS), Institut sur la Nutrition et les Aliments Fonctionnels (INAF), Université Laval, Québec, QC G1V 0A6, Canada; 2École de Nutrition, Faculté des Sciences de l’Agriculture et de l’Alimentation, Université Laval, Québec, QC G1V 0A6, Canada; 3Institut National de Santé Publique du Québec, Québec, QC G1V 5B, Canada

**Keywords:** Canada’s Food Guide, diet quality, diet cost, food price, Healthy Eating Food Index-2019, sustainability, sustainable diet

## Abstract

The aim of this study was to assess the association between daily diet costs and the Healthy Eating Food Index (HEFI)-2019, an index that reflects the alignment of dietary patterns to recommendations on healthy food choices in the 2019 Canada’s Food Guide (CFG). Dietary intake data from 24 h recalls, completed between 2015 and 2017, of 1147 French-speaking participants of the web-based multicenter cross-sectional PRÉDicteurs Individuels, Sociaux et Environnementaux (PREDISE) study in Quebec were used. Diet costs were calculated from dietary recall data using a Quebec-specific 2015–2016 Nielsen food price database. Usual dietary intakes and diet costs were estimated using the National Cancer Institute’s multivariate method. Linear regression models were used to evaluate associations between diet costs and HEFI-2019 scores. When standardized for energy intake, a higher HEFI-2019 score (75th vs. 25th percentiles) was associated with a 1.09 $CAD higher daily diet cost (95% CI, 0.73 to 1.45). This positive association was consistent among different sociodemographic subgroups based on sex, age, education, household income, and administrative region of residence. A higher daily diet cost was associated with a higher HEFI-2019 score for the Vegetables and fruits, Beverage, Grain foods ratio, Fatty acids ratio, Saturated fats, and Free sugars components, but with a lower score for the Sodium component. These results suggest that for a given amount of calories, a greater adherence to the 2019 CFG recommendations on healthy food choices is associated with an increased daily diet cost. This highlights the challenge of conciliating affordability and healthfulness when developing national dietary guidelines in the context of diet sustainability.

## 1. Introduction

International organizations are calling for a transition toward more sustainable food systems, which entail important changes in eating habits at the population level [1,2]. The Food and Agriculture Organization of the United Nations (FAO) has defined sustainable diets as “those with low environmental impacts that contribute to food and nutrition security and to healthy life for present and future generations”. The FAO also states that “Sustainable diets are protective and respectful of biodiversity and ecosystems, culturally acceptable, accessible, economically fair and affordable, nutritionally adequate, safe, and healthy, while optimizing natural and human resources” [3]. Although studies on sustainable dietary patterns tend to focus on the environmental impacts of foods [4], the economic accessibility of healthy foods and whole diets is a major consideration for sustainability. This is not a trivial consideration, because food cost has been identified as one of the main determinants of dietary choices by consumers [5,6]. 

Previous studies have found that foods considered healthier tend to cost more than less healthy foods [7,8]. For example, fruits and vegetables cost more per calorie than foods that are more energy dense, such as refined grains, fats and sweets [9,10,11]. Consequently, evidence suggests that healthy dietary patterns are more expensive than unhealthy diets [8]. For example, diet quality assessed by multiple indexes such as the Mediterranean Diet score [12], the Diet Approach to Stop hypertension (DASH) score [13,14], the Healthy Eating Index (HEI) [15,16] and the Alternative Healthy Eating Index (AHEI) [16,17] has previously been positively associated with higher diet costs at the individual level. In this context, low adherence to healthy dietary patterns may be explained in part by economic constraints. This hypothesis is supported by a meta-analysis that demonstrated that lower-nutritional-value foods and diets tend to be selected by groups with lower socioeconomic status [18]. 

The revised version of the Canada’s Food Guide (CFG) issued in 2019 promotes healthy food choices as well as healthy eating habits [19]. In addition to previous recommendations on vegetables and fruits, whole grains, saturated fat, sodium, and sugar, the new dietary guidelines for Canadian consumers also put more emphasis on plant-based foods as well as on healthy beverages. We have recently developed and evaluated the Healthy Eating Food Index (HEFI)-2019 to assess the alignment of dietary patterns to recommendations on healthy food choices in the 2019 CFG [20,21]. This new index provides an opportunity to examine how the alignment to recommendations on healthy food choices in Canada influences the cost of the daily diet. From the perspective of the economic accessibility of sustainable dietary patterns, the aim of this study was to examine the association between daily diet costs and the HEFI-2019 in a sample of adults from the province of Quebec. We hypothesized that alignment with CFG recommendations on healthy food choices increases the daily diet cost, with variable impacts among sociodemographic groups.

## 2. Materials and Methods

### 2.1. Study Design and Population

The data used for these analyses were from the web-based multicenter cross-sectional PRÉDicteurs Individuels, Sociaux et Environnementaux (PREDISE) study. The aim of the PREDISE study was to document associations between individual, social and environmental factors, and the adherence to dietary guidelines. The complete methods of the PREDISE study have been described elsewhere [22]. Briefly, participants were recruited between August 2015 and April 2017 through a survey firm in five administrative regions of the province of Quebec: Capitale-Nationale/Chaudière-Appalaches, Estrie, Mauricie, Montreal and Saguenay-Lac-St-Jean. Stratified sampling was used to obtain an age- and sex-representative sample of French-speaking adults aged 18 to 65 years for the five administrative regions. Pregnant or lactating women were excluded from the study. Participants completed online questionnaires over a three-week period regarding sociodemographic data as well as three unannounced validated web-based 24 h dietary recalls (R24W) [23,24]. Among the 1849 participants who met the inclusion criteria and gave their written consent, 1147 completed at least one 24 h recall and were included in the study (see Appendix A). The project was conducted in accordance with the Declaration of Helsinki and was approved by the Research Ethics Committees of Université Laval (ethics number: 2014-271), Centre hospitalier universitaire de Sherbrooke (ethics number: MP-31-2015-997), Montreal Clinical Research Institute (ethics number: 2015-02), and Université du Québec à Trois-Rivières (ethics number: 15-2009-07.13).

### 2.2. Dietary Intake Assessment

Dietary intakes were assessed using three unannounced R24W, including one on a weekend day, over a three-week period. The R24W is a web-based automated self-administered instrument in which participants are asked to report all foods they consumed the day before. Complete procedures about the development and validation of the R24W have been described previously [23,24]. Mixed dishes in the R24W can be broken down into individual foods from the Canadian Nutrient File (CNF). The nutrient value of foods and mixed dishes reported in the R24W is generated automatically using the 2015 CNF. Each of the foods reported is also linked to a Bureau of Nutritional Science (BNS) food group (*n* = 180) of the 2015 CNF to allow the calculation of diet costs. 

### 2.3. Healthy Eating Food Index-2019

The HEFI-2019 was calculated using dietary intake data obtained from 24 h recalls. The HEFI-2019 includes 10 components, five of which are based on foods (Vegetables and fruits, Whole grain foods, Grain foods ratio, Protein foods, Plant-based protein foods), one of which is based on beverages (Beverages), and four of which are based on nutrients (Fatty acids ratio, Saturated fats, Free sugars, Sodium). Each component is scored on a 5- to 20-point scale, for a total maximum score of 80 (Appendix A, see full description of the development and evaluation of the HEFI-2019 [20,21]). Higher HEFI-2019 scores indicate a better alignment of a dietary pattern with recommendations on healthy food choices in the 2019 CFG. 

### 2.4. Food Price Data

Retail food prices used were compiled by the Nielsen company by scanning the barcodes of products purchased at cash registers. These data were originally collected for market research purposes and include annual purchase data in Canadian dollars ($CAD) and kilograms (kg) made at the three largest grocery chains in Quebec (Loblaw, Sobeys, Metro), as well as some big box stores (Walmart, Target, Zellers) during the 2015–2016 fiscal year. Therefore, the retail food prices used correspond to the same data collection period of the PREDISE study, and reflect the average prices paid by consumers across the Province of Quebec during that period. 

A food price database was created by calculating a standard price for each BNS food group (*n* = 180) of the 2015 CNF. The food price for each food was calculated by dividing the total cost of foods purchased at the predetermined grocery chains and big box stores in Quebec in 2015–2016 by the total amount of foods purchased within each of the BNS food groups. The food price of each BNS food group was weighted using the volume of purchase of each food within each group. For example, the average price of the BNS food group 36E-Carrots reflected the combined price of fresh, canned and frozen carrots, as weighted according to the relative quantities of each food category purchased at the three grocery chains and big box stores in 2015–2016. For several BNS food groups (*n* = 47), it was not possible to calculate a standard price using the Nielsen database. Examples are Sugar substitutes, Wine, Beer, Butter, Cottage cheese, Coffee, Table cream, Vegetable oils and Meal replacements. For these, different scenarios were used (See Appendix A). First, missing prices were obtained using statistical data from the Government of Canada [25] or Quebec [26] for the corresponding dates (i.e., the 2015–2016 fiscal year). Second, prices of similar foods were compared using a supermarket website and the missing prices were replaced with the price of comparable foods from the Nielsen database, the Statistics Canada database, or the Institut de la statistique du Québec database, with the application of a factor if necessary. For example, the price of lamb was calculated as 1.7× the price of beef based on data from Nielsen. For alcoholic beverages, prices were obtained from the 2016 annual report of the Société des alcools du Québec [27]. For the remaining BNS food groups *(n* = 6), no price was estimated, since these food items were either free (e.g., water) or had little impact on the total diet cost because their consumption is typically very low (e.g., Babyfood products, Spices, Seasonings, Game meat). The price of mixed dishes was calculated by adding the costs of its individual food constituents. Food and mixed dish prices were then adjusted for material loss, using data from the CNF [28], and for food preparation to account for moisture and fat loss as well as cooking gains [29]. Thus, this allowed the calculation of a price per edible portion expressed in dollars per kilogram for each of the 180 BNS food groups. 

### 2.5. Diet Cost

The diet cost was calculated for each participant and each 24 h recall using reported food and beverage intake data by multiplying the amount of each food or beverage reported expressed in kilograms by the corresponding BNS food group price per kilogram. The cost of each food item consumed was then summed, yielding a daily diet cost for each of the three 24 h recalls. For descriptive purposes, an energy-adjusted daily diet cost, reported per 2500 kcal, was also calculated. 

### 2.6. Statistical Analyses

Analyses were performed in SAS Studio (version 3.81 SAS Institute, Cary, NC, USA), and figures were generated in R Studio (version 2022.02.0; R Foundation for Statistical Computing, Vienna, Austria). SURVEY procedures were used to account for the stratified design of the study when applicable. Because the PREDISE final sample size was larger than originally aimed for, balancing weights were used to ensure sex and age representativeness in each administrative region. Missing sociodemographic characteristics were imputed (education (*n* = 60), income (*n* = 159)) using the fractional imputation method. For descriptive purposes, the mean daily diet cost and energy-adjusted daily diet cost in the overall population and among predetermined sociodemographic subgroups based on sex (men and women), age (18 to 34 y, 35 to 49 y, 50 to 65 y), education (none, high school, trade/CEGEP/university), household income (<30,000 $CAD, 30,000 to <60,000 $CAD, 60,000 to <90,000 $CAD, ≥90,000 $CAD) and administrative region was estimated using the SURVEYMEANS procedure. The mean HEFI-2019 score and component scores were estimated for the entire sample as well as sociodemographic subgroups using the population ratio method for descriptive purposes [30]. Standard errors and 95% CI were estimated using 200 bootstrap resamples. Distribution of usual (i.e., long term) food and nutrient intakes as well as of daily diet costs were estimated using the National Cancer Institute (NCI)’s multivariate Markov Chain Monte Carlo method [31]. This method attenuates the impact of within-individual random errors using regression calibration of data from repeated 24 h recalls. The model was stratified by sex to better reflect within-individual random dietary intake variations. Covariates included indicators for the sequence of 24 h recalls (i.e., first, second or third recall) and the day of the week (i.e., weekdays vs. weekend). Additional covariates were included to output subgroup distributions according to age, education, household income and administrative region. Foods were considered to be consumed episodically when 10% or more of the population did not report consumption on the first dietary recall. Accordingly, whole grain foods, plant-based protein foods and some beverages (i.e., sugary drinks, artificially sweetened beverages, vegetable and fruit juices, sweetened milk and plant-based beverages, alcohol, unsweetened milk, and unsweetened soy beverages) were considered episodic foods in the NCI’s multivariate models. All remaining foods and nutrients were considered to be consumed daily. The diet cost, which was nonzero for all participants for each of the three 24 h recalls, was also considered as a “daily” variable in the NCI’s multivariate method. Usual intakes and usual diet costs were estimated from 500 pseudo-individuals generated during the Monte Carlo simulation step and were pooled within each sex stratum. The HEFI-2019 score and component scores were calculated from estimated usual intakes among pseudo-individuals. 

The association between diet costs and energy intake was assessed using a regression model with 4-knot restricted cubic splines (RCS) (5th, 35th, 65th, and 95th percentiles of energy intake). Regression models with RCS were also used to assess associations between diet costs and the HEFI-2019 score and component scores in the overall sample and in predetermined sociodemographic subgroups. Knots at the 5th, 35th, 65th and 95th percentiles of the HEFI-2019 score and component scores were used, except for the Sugar component, for which knots at the 10th, 50th and 90th percentiles were used to better account for its skewed distribution. For the Protein foods component, a linear regression model without RCS was used. Effect sizes were estimated using the difference in the diet cost between the 75th and 25th percentiles of the HEFI-2019 score and component scores distribution, with and without adjustment for daily energy intake. Standard errors and 95% CI were estimated using 200 bootstrap resamples. 

## 3. Results

Characteristics of study participants, energy-adjusted daily diet cost, and HEFI-2019 score are presented in Table 1. The sample was representative of the age and sex of adults in each of the five administrative regions after balancing weights were applied to the data. The study sample included 50.2% women, and 44.7% of participants had a university degree. The total annual household income was >90,000 $CAD among 34.8% of participants. The HEFI-2019 score in the total sample was 45.1 points (95% CI, 44.3 to 45.9). Being a female, being older and having a higher degree of education were associated with a higher HEFI-2019 score. The mean energy-adjusted daily diet cost was 12.72 $CAD/2500 kcal (95% CI, 12.53 to 12.92) in the total sample. In general, being older and having a higher household income were associated with a higher energy-adjusted daily diet cost. The daily diet cost unadjusted for energy is presented in Appendix A.

Figure 1 illustrates the association between estimated daily diet costs and the HEFI-2019 score. The estimated cost difference between high (75th percentile) and low (25th percentile) HEFI-2019 scores (difference of 13.5 points) is also shown. As expected, there was a strong positive association between daily diet costs and total energy intake (r = 0.73, 95% CI, 0.69 to 0.77, data not shown). Before adjustment for total energy intake, there was no difference in the diet cost between the 75th and 25th percentiles of the HEFI-2019 score distribution (0.35 $CAD/day, 95% CI, −0.23 to 0.93). After adjustment for total energy intake, the estimated diet cost at the 75th percentile of the HEFI-2019 score distribution was 1.09 $CAD/day (95% CI, 0.73 to 1.45) greater than at the 25th percentile. Thus, from here on in the present paper, analyses are adjusted for total energy intake to account for the strong association between cost and energy intake.

As shown in Table 2, the significance and magnitude of the diet cost difference between the 75th and 25th percentiles of the HEFI-2019 score distribution were consistent among all sociodemographic subgroups defined according to sex, age, education, household income and administrative region. 

Figure 2 shows the association between estimated daily diet costs and HEFI-2019 component scores. The positive association between daily diet costs and the total HEFI-2019 score was attributable to the following components: Vegetables and fruits (difference in daily diet costs, 75th vs. 25th percentiles: 1.13 $CAD/day, 95% CI 0.74 to 1.52), Grain foods ratio (0.71 $CAD/day, 95% CI 0.32 to 1.09), Beverages (0.59 $CAD/day, 95% CI 0.20 to 0.99), Fatty acids ratio (0.41$CAD/day, 95% CI 0.00 to 0.81), Saturated fats (0.54 $CAD/day, 95% CI 0.07 to 1.00) and Free sugars (1.99 $CAD/day, 95% CI 1.62 to 2.36). In contrast, high vs. low scores for the Sodium component was associated with a lower daily diet cost (−0.59 $CAD/day, 95% CI, −1.03 to −0.14). No difference in the daily diet cost was observed between high and low scores for the Whole grain foods, Protein foods and Plant-based protein foods components of the HEFI-2019. Of note, the sum of the differences in the estimated daily diet cost associated with each component does not add up to the difference in the daily diet cost for the HEFI-2019 total score, as each component does not have the same weight in the HEFI-2019 scoring algorithm.

## 4. Discussion

The aim of this study was to document for the first time the association between daily diet costs and the alignment of dietary patterns to recommendations on healthy food choices in the 2019-released CFG, as measured by the HEFI-2019, among adults in the province of Quebec. The findings confirmed that for a given amount of calories, a greater degree of alignment to recommendations on healthy food choices increases the daily diet cost. Contrary to our hypothesis, the positive association between diet quality according to Canadian recommendations and estimated diet costs was consistent among several sociodemographic subgroups based on sex, age, education, household income and administrative region. Data further suggested that the increase in daily diet costs was primarily driven by the Vegetables and fruits, Grain foods ratio, Beverages, Fatty acids ratio, Saturated fats, and Free sugars components of the HEFI-2019, while the Sodium component was inversely associated with estimated daily diet costs. 

The findings from our study are consistent with data from previous studies in other jurisdictions suggesting that dietary patterns that align with dietary guidelines cost more than lower-quality diets. For example, a 2018 study found that energy-standardized dietary patterns meeting the UK recommendations for fruits and vegetables, oily fish, non-milk extrinsic sugars, fats, saturated fats, and salt were estimated to cost 3 to 17% more than dietary patterns that failed to meet these recommendations [13]. The authors also reported that dietary patterns meeting multiple recommendations simultaneously cost 29% more than patterns that met none of the recommendations [13]. Two other studies conducted in the USA by Rehm and al. [15,32] examined the association between diet costs and adherence to the 2005 and 2010 US dietary guidelines, measured using the HEI-2005 and -2010, respectively. Both studies found that a better alignment with the US dietary guidelines was associated with a higher energy-adjusted diet cost. A meta-analysis published in 2013 also observed that healthy food-based dietary patterns, assessed using different food quality indexes, cost on average USD 1.48/day and USD 1.54/2000 kcal more than less healthy patterns [8]. Hence, the increased diet cost associated with a greater degree of alignment of dietary patterns with the 2019 CFG recommendations on healthy food choices is consistent with the broader evidence that healthier dietary patterns are generally more expensive. 

Our findings did not support our hypothesis that the association between the HEFI-2019 and diet costs differs by sociodemographic characteristics. Regardless of sex, age, education, household income or administrative region, a better alignment of dietary patterns to recommendations on healthy food choices in the 2019 CFG was associated with a higher daily diet cost. These findings are only partially coherent with previous studies. For example, Rhem et al. [15,32] also found a positive association between diet costs and the HEI-2005 and -2010 regardless of sex, although the association was stronger among women than among men. A positive association between the monetary value of diet and the HEI-2010 among subgroups of individuals based on sex and poverty status was observed by Beydoun et al. [33]. However, the authors found a stronger association among women and individuals above poverty status, in contrast with our results, where no difference was observed. 

Consistent with previous findings [34], the Vegetables and fruits component of the HEFI-2019 was strongly associated with daily diet costs. This is unsurprising, considering that this component accounts for up to 25% of the HEFI-2019 score (20 points/80). These findings are also consistent with well-established evidence that low-energy-density foods that are also more nutrient rich, such as vegetables and fruits, cost more per calorie than energy-dense foods that are high in sugars and fats, such as refined grains, sweets and snacks [9,11,35,36]. The Sodium component of the HEFI-2019 was inversely associated with diet cost, consistent with data from other studies [15,32]. This may be explained by the fact that sodium is ubiquitously present in a wide variety of foods, including “healthier” foods such as whole grain products and plant-based protein foods [37], which had null effects on diet costs. Somewhat unexpectedly, a higher score for the Plant-based protein component of the HEFI-2019 had no incidence on the estimated daily diet cost. However, participants with higher intakes of plant-based protein also consumed more nutrient-rich foods such as vegetables and fruits (data not shown), possibly cancelling out the impact of lower cost plant-based protein on total daily diet costs. This reinforces the hypothesis that promoting plant-based protein foods in the context of food and diet sustainability is an effective way to improve diet quality with minimal impact on diet costs [38].

The increased diet cost associated with a greater alignment with current Canadian dietary guidelines on healthy food choices may be an important barrier for consumers in adopting healthy dietary patterns. Indeed, an increase of 1.09 $CAD per day represents a 9% increase in the average daily diet cost in this fairly highly educated population. Over a one-year period, this would translate into an estimated increase of 397.85 $CAD for one adult, a major barrier to sustainability, particularly among low-income subgroups. To mitigate this, pricing measures such as subsidies have been shown to be an effective strategy to improve consumption of healthy foods [39,40]. A 2017 meta-analysis reported that every 10% decrease in healthy food prices was associated with a 12% increase in their purchase and consumption [41]. It should be noted that the food price data used in our study are from 2015 to match the collection of the dietary intake data. The current food prices at the time of this analysis (2022) have increased in Québec and elsewhere in the world due, among other things, to COVID-19 pandemic-related inflation [42]. Consumers are therefore facing even greater challenges in dealing with the important increase in the cost of the grocery basket [43,44]. Food pricing strategies and policies in various forms, such as discounts, coupons, cash rebates, and subsidies to make healthy dietary patterns more affordable and sustainable, particularly to the vulnerable segments of the population, may be even more important today to attenuate the risk of further diet-related health inequalities.

This study has several strengths, including the use of an age- and sex-representative sample of the French-speaking adult population of the province of Quebec. Although self-reported dietary intake data are affected by errors [45,46,47], the use of the NCI’s multivariate method is a major strength, as it allowed us to account for random errors and generate data of usual dietary intakes as well as of daily diet costs, as opposed to reporting data on “any given day”. Another strength of our study is the use of a food price database of actual costs paid by consumers during the study period rather than food prices collected from a convenience sample. Some limitations must also be addressed. Although the sample is representative of sex and age distribution among French-speaking residents of the province of Quebec, caution is advised when generalizing the results to other populations, considering the slightly higher education and income levels of the sample. Second, the Nielsen food price database used represents the average price paid by consumer across the Province of Quebec. Food prices are not available by type of store, season, geographic location, or other demographic factors. They also do not represent the lowest price available. Thus, caution should be exercised when interpreting subgroup analyses as well as absolute diet cost differences between groups. Third, by using food price data from Nielsen, we worked under the assumption that all foods and beverages were bought from grocery or big box stores. Food consumed at home and food consumed away from home (i.e., at restaurants and other outlets) were not distinguished. Food waste was also not considered. 

## 5. Conclusions

In conclusion, the present study provides evidence that, for a given amount of calories, dietary patterns that better align to the 2019 CFG recommendations on healthy food choices increase the daily diet cost. In the context of sustainability, these findings are particularly relevant, and highlight the challenge of conciliating affordability and healthfulness when developing national dietary guidelines. These findings may also be used by policymakers as a starting point to make healthy food choices more affordable in the province of Quebec and in Canada.

## Figures and Tables

**Figure 1 nutrients-14-03818-f001:**
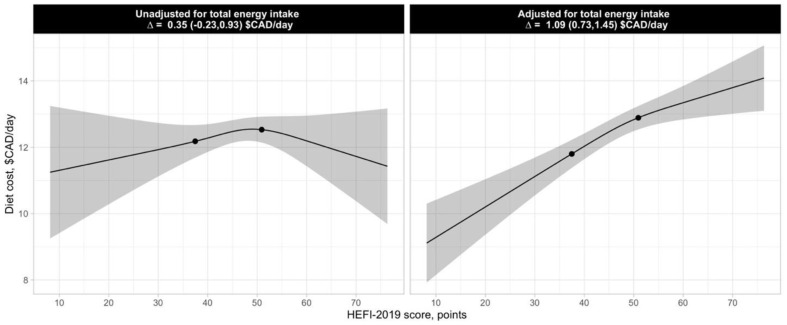
Linear regression of HEFI-2019 scores and estimated daily diet costs in 1147 French-speaking adults from Quebec. A higher HEFI-2019 score indicates a greater adherence to recommendations on healthy food choices in the 2019 CFG. The 25th and 75th percentiles of the HEFI-2019 score distribution are represented by the black dots on the regression line. The estimates presented correspond to the daily diet cost difference between the 75th and 25th percentiles of the HEFI-2019 score distribution. The 95% CI of the regression is represented by the shaded area. The left panel presents the linear regression unadjusted for total energy intake, while the right panel presents the linear regression adjusted for total energy intake. Estimated diet costs and dietary intake data were obtained using the National Cancer Institute’s multivariate method and reflect usual (i.e., long-term) intakes and costs; 95% CI was estimated using 200 bootstrap resamples. CAD, Canadian dollars; HEFI-2019, Healthy Eating Food Index-2019.

**Figure 2 nutrients-14-03818-f002:**
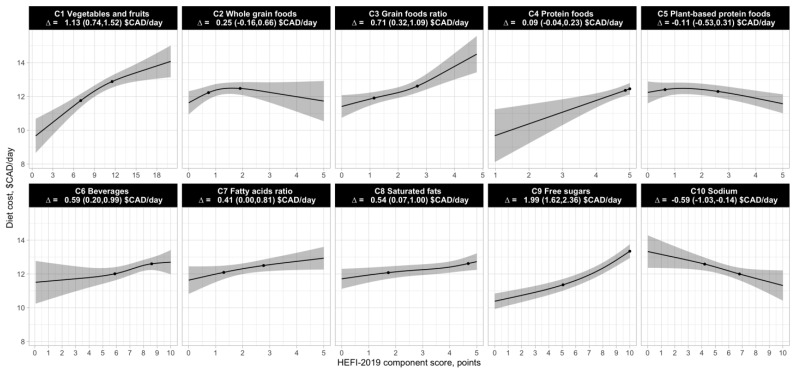
Linear regression of HEFI-2019 component scores and estimated daily diet costs adjusted for total energy intake in 1147 French-speaking adults from Quebec. Higher HEFI-2019 component scores indicate a greater adherence to individual recommendations in the 2019 CFG. The 25th and 75th percentiles of the HEFI-2019 component score distribution are represent by the black dots on the regression line. The estimates presented correspond to the usual daily diet cost difference between the 75th and 25th percentiles of the HEFI-2019 component score distribution. The 95% CI of the regression is represented by the shaded area. Estimated diet costs and dietary intake data were obtained using the National Cancer Institute’s multivariate method and reflect usual (i.e., long-term) intakes and costs. The 95% CI was estimated using 200 bootstrap resamples. CAD, Canadian dollars; HEFI-2019, Healthy Eating Food Index-2019.

**Table 1 nutrients-14-03818-t001:** Participants’ sociodemographic characteristics, energy-adjusted daily diet cost and Healthy Eating Food Index-2019 score ^1^.

	Participants (*n* = 1147)	Energy-Adjusted Daily Diet Cost $CAD/2500 kcal (95% CI) ^2^	HEFI-2019 Score (95% CI) ^3^
**All**		12.72 (12.53, 12.92)	45.1 (44.3, 45.9)
**Sex**			
Women	576 (50.2)	12.75 (12.48, 13.02)	47.6 (46.7, 48.6)
Men	571 (49.8)	12.70 (12.42, 12.97)	43.0 (41.9, 44.1)
**Age**			
18–34 y	408 (35.6)	12.00 (11.68, 12.31)	42.9 (41.6, 44.2)
35–49 y	338 (29.5)	12.72 (12.34, 13.10)	45.6 (44.1, 47.1)
50–65 y	400 (34.9)	13.46 (13.14, 13.79)	47.0 (45.8, 48.3)
**Education ^4^**			
None, high school or trade diploma	282 (24.6)	12.52 (12.09, 12.94)	42.3 (40.6, 44.0)
CEGEP	351 (30.6)	12.84 (12.47, 13.20)	44.7 (43.3, 46.1)
University	513 (44.7)	12.76 (12.48, 13.03)	46.9 (45.8, 48.1)
**Household income ^4^**			
<30,000 $CAD	203 (17.7)	11.72 (11.26, 12.18)	42.0 (40.1, 43.9)
30,000 to <60,000 $CAD	328 (28.6)	12.75 (12.37, 13.13)	44.9 (43.4, 46.4)
60,000 to <90,000 $CAD	217 (19.0)	12.87 (12.44, 13.30)	46.8 (45.0, 48.7)
≥90,000 $CAD	399 (34.8)	13.13 (12.80, 13.45)	45.9 (44.7, 47.1)
**Administrative region**			
Capitale-Nationale/ Chaudière-Appalaches	435 (37.9)	13.04 (12.70, 13.37)	45.1 (43.8, 46.5)
Estrie	110 (9.6)	12.60 (12.08, 13.13)	45.8 (43.2, 48.4)
Mauricie	99 (8.6)	13.31 (12.56, 14.06)	44.1 (41.3, 46.8)
Montreal	397 (34.6)	12.37 (12.07, 12.67)	46.6 (45.3, 47.9)
Saguenay-Lac St-Jean	107 (9.3)	12.35 (11.73, 12.96)	40.2 (38.1, 42.4)

^1^ CAD, Canadian dollars; CEGEP, Collège d’Enseignement Général et Professionnel; HEFI-2019, Healthy Eating Food Index-2019. ^2^ Because of their descriptive nature, these mean costs are not based on the National Cancer Institute’s multivariate method. ^3^ HEFI-2019 scores estimated using the population ratio method [30]; 95% CI calculated using 200 bootstrap resamples. ^4^ Missing sociodemographic characteristics have been imputed. See the Methods section for details.

**Table 2 nutrients-14-03818-t002:** Estimated diet cost difference between high vs. low HEFI-2019 scores in 1147 French-speaking adults from Quebec according to sociodemographic characteristics ^1^.

	Diet Cost Difference Estimate ($CAD/Day) 75th vs. 25th Percentiles of the HEFI-2019 Score Distribution	95% CI ^2^
**Sex**		
Women	1.02	0.65–1.40
Men	1.18	0.66–1.69
**Age**		
18–34 y	1.03	0.69–1.37
35–49 y	1.02	0.64–1.41
50–65 y	0.93	0.54–1.31
**Education ^3^**		
None, high school or trade diploma	1.16	0.72–1.59
CEGEP	1.04	0.68–1.41
University	1.05	0.67–1.42
**Household income ^3^**		
<30,000 $CAD	1.12	0.77–1.47
30,000 to <60,000 $CAD	0.90	0.49–1.31
60,000 to <90,000 $CAD	0.89	0.50–1.27
≥90,000 $CAD	1.13	0.73–1.52
**Administrative region**		
Capitale-Nationale/ Chaudière-Appalaches	1.10	0.71–1.49
Estrie	0.99	0.50–1.47
Mauricie	1.20	0.73–1.66
Montreal	1.14	0.81–1.47
Saguenay-Lac St-Jean	1.26	0.84–1.69

^1^ HEFI-2019 scores and estimated diet costs based on usual intakes and costs obtained with the National Cancer Institute’s multivariate method [31]. Results are adjusted for total energy intake. CAD, Canadian dollars; CEGEP, Collège d’Enseignement Général et Professionnel; HEFI-2019, Healthy Eating Food Index-2019. ^2^ 95% CI estimated using 200 bootstrap resamples. ^3^ Missing sociodemographic characteristics have been imputed. See the Methods section for details.

## Data Availability

The data presented in this study are available on request from the corresponding author.

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
