# Peer review of "Adhering to Canada’s Food Guide Recommendations on Healthy Food Choices Increases the Daily Diet Cost: Insights from the PREDISE Study"

_nutrients, 2022, doi:10.3390/nu14183818_

Round 1
Reviewer 1 Report
Thank you for the opportunity to review this manuscript. I really enjoyed reading it and think it is of substantial interest at present due to rising food prices affecting the world. This novel analysis demonstrated that increasing compliance to healthy eating recommendations is associated with substantially increased cost of diet, that is particularly related to fruits and vegetables.
I only have very minor comments which could be addressed by the authors, but generally it is a well written manuscript.
First, regarding your hypothesis of there being variable impacts among sociodemographic groups. This is explained in the discussion further down, but the first time reader could be better oriented within the introduction as to why the authors consider there could be differences between some sociodemographic groups - ie. what are the groups at potentially higher risk and why?
In the limitations section, could the authors comment on the limitations of the study sample and whether the findings are generalisable or not to other populations?
L342-3 - I feel this is is an important consideration that warrants further discussion. Can the authors comment on what the implications of rising food prices could be and make recommendations for research/policy that should be prioritised in the context of rising food prices? Are your study findings relevant to the context of 2022?
Reviewer 2 Report
This a well written, interesting paper that contributes to the literature on the important topic of healthy, affordable and sustainable diets in developed countries.
Some specific comments/suggestions below:
Materials and Methods
2.3 Food price data. The approach was to apply a price per kilo for each food group rather than individual foods. This seems sensible if foods within a food group are very similar. It would be helpful to see all 180 food groups in Table S1.
Results
Given the association between diet cost and energy intake, suggest that Table 1 presents the energy adjusted diet cost or both. Figure 1 could come first.
Figure 1. It is really difficult to spot the estimates at the top of figure. Suggest black background as in Figure 2.
Line 240. Clarify what is meant by ‘consistent’. Consistent in that always a significant positive association rather than that the cost difference is consistent?
Discussion
The results show a higher score is associated with higher cost for free sugars and saturated fat but inverse for sodium. Can the authors offer any reason why this might be?
As a limitation, authors should mention any bias associated with the diet data collected/dietary assessment method.
Reviewer 3 Report
Thank you for giving me an opportunity to review your manuscript. I have a few comments.
1. The dietary data in on what foods people actually ate (lines 146-150), and not foods purchased. The foods people reported eating will be sandwiches, soups, casseroles, prepared rice dishes, salads, etc. How did the authors link the price of food (food ingredients) as purchased to foods as prepared and consumed? Most food consumed are composed of more than one food group. It is not clear how these mixed foods were treated. This information linking food purchased to food mixtures (food with more thank one food group) consumed is not explained. Without this information, it is not possible to assess whether the findings reported are correct or have merit. Lines 117 to 114 do not explain this. The examples given in the manuscript are for simple food items that can be purchased as such. There is no mention of prepared food item that one encounters in a dietary recall.
2. Please list the components of HEFI and the scoring system for each of the component in a table. Referring a reader to a publication doesn’t help, because your study uses HEFI as the basis for evaluating diet quality.
3. Table 1. Females have a higher HEFI score than males. However, the difference in the score is just 5 points. Which components are responsible for this difference? What is the practical significance of this 5 point difference? To me, it looks like the mean HEFI scores seem to be between 40 and 47 for all demographic group. The maximum score is 80. Hence, it does not look like most groups or having an average diet. Also men’s HEFI score is low but diet cost is higher than women’s. Is it because men eat more?
4. Table 1. Please add a column on mean energy (calories) intakes of these demographic groups.
5. Lines 316-332. Is it possible fruit and vegetables cost most not because they are more nutritious, but because they are highly perishables when compared to gain-based products?
6. Lines 344. Not sure about the strength of the study since it is not clear how the price/cost of foods reported consumed in the dietary recall is not explained in the manuscript.
